# Clinical Trial Studies of Antipsychotics during Symptomatic Presentations of Agitation and/or Psychosis in Alzheimer's Dementia: A Systematic Review

Haider Qasim *, Maree Donna Simpson and Jennifer L. Cox

School of Medical Sciences and Dentistry, Charles Sturt University, Orange Campus, Orange, NSW 2800, Australia
* Correspondence: hqasim@csu.edu.au

**Abstract:** Aggressive behaviors of people with dementia pose a significant challenge to employees in nursing homes and aged care facilities. Aggressive behavior is a result of psychomotor agitation in dementia (BPSD). Globally, psychotropic interventions are the preferred treatment for BPSD. However, it is still unclear which psychotropic should be prescribed. The purpose of this systematic review is to compare pharmacological interventions for psychomotor agitation and psychosis symptoms. Method: The studies were extracted from databases, such as PubMed, OVID, and Cochrane, with a date restriction from 2000 to present, and in English. PRISMA steps were used to refine the extracted data. The RCTs extracted for this systematic review compared active ingredient medications to one another or to a placebo. Results: PRISMA was used to assess all selected trials comprehensively. Four trials are being conducted on quetiapine, two on haloperidol, one on olanzapine, three on risperidone, one on brexpiprazole, one on pimavanserin, and two on aripiprazole. Compared to typical antipsychotics, quetiapine showed tolerable adverse effects and did not worsen parkinsonism. Psychosis symptoms and behavioral improvements can be improved with haloperidol. Among elderly patients with psychosis, risperidone reduces angriness, paranoia, and aggression, as well as improves global functioning. As compared with other antipsychotics, aripiprazole provides a lower risk of adverse effects and demonstrated improvement in agitation, anxiety, and depression associated with psychosis. While olanzapine improves hostile suspiciousness, hallucinations, aggression, mistrust, and uncooperativeness, it worsens depression symptoms. Psychosis was treated effectively with pimavanserin without adverse effects on motor functions. Psychosis symptoms are well tolerated by brexpiprazole, but insomnia, headache, and urinary tract infections are common side effects. Conclusions: In this systematic review, we provide an overview of how to choose the correct antipsychotics and dosages for the management of BPSD and emphasize the importance of safe and conservative use of these drugs.

**Keywords:** antipsychotics; Alzheimer's dementia; psychosis; agitation; clinical trials

## 1. Introduction

Behavioral and psychological symptoms of Alzheimer's disease are non-cognitive neuropsychiatric symptoms that occur in older people with cognitive impairment and represent a heterogeneous group of psychopathological signs. Behavioral disorders are characterized by loud vocalizations, pacing, aggression, hoarding, and walking about, while psychological disorders are characterized by apathy, depression symptoms, anxiety, delusion, and hallucinations [1]. It is a significant challenge for employees of aged care facilities and nursing homes to deal with violence perpetrated by demented people at work [2]. Behavior and psychological disorders (such as agitation and psychosis) result in violence in dementia patients [1,2]. Approximately 90% of individuals with dementia experience this condition over their lifetime [3]. In addition, patients may experience abnormal motor behaviors, apathy, sleep disturbances, agitation, anxiety, depression,

irritability, psychosis, delusions, hallucinations, and changes in appetite [4]. Symptoms associated with these conditions have been identified as a dementia risk factor [2–5], especially in conjunction with psychotic symptoms. As a result, BPSD can negatively affect the quality of life, illness, treatments, family and professional relationships, and caregiver burdens. Thus, it is essential to customize the treatment of BPSD based on factors such as effective pain management, proper management of somatic diseases, and the optimal use of non-pharmacological interventions and pharmacotherapy. In most cases, behavioral management, environmental modification, sensory interventions, and social interaction groups can be used as non-pharmacological interventions for treating agitation [6,7]. There are many instances in which behavioral interventions alone are not sufficient to control agitation episodes, necessitating medication treatment in some cases [8,9]. The primary goal of pharmacological intervention is to rapidly calm the agitated patient [9]. Verbal de-escalation and environmental modification techniques should be used in conjunction with the treatment. Where possible, the selection of medication needs to be given firstly as a monotherapy, with rapid onset of action a desirable feature [9,10]. Despite the fact that only antipsychotics and benzodiazepines are effective in treating agitation, this is of concern. These treatments, however, can have serious adverse effects, especially in patients who are vulnerable, such as those suffering from dementia [11]. Further, these medications have been shown to contribute to metabolic and cardiovascular diseases, as well as cognitive decline [12]. For these reasons, clinicians are still working to identify novel therapeutic options for dementia, focusing on symptomatic treatments for agitation and psychosis [13,14].

## 2. Research Questions

Are atypical antipsychotic treatments a safe and effective option for the management of agitation and psychosis related symptoms in older persons diagnosed with Alzheimer's dementia? How can the outcomes of this review be translated to application in clinical practice and is there enough information to form a treatment guideline to assist physicians with customized prescribing practices?

### 2.1. Methods

In this systematic review, we examined evidence from RCTs, placebo-controlled trials comparing antipsychotics and/or placebos for BPSD patients who had been treated for agitation or psychosis. A variety of databases were searched, including Embase via Ovid, PubMed, Scopus, Cochrane Central Register of Controlled Trials, CINAHL plus (EBSCOhost), Epistemonikos, Ovid MEDLINE, Web of Science core collection, International pharmaceutical abstracts (ProQuest), MedlinePlus, and PsychINFO.

We also researched primary studies used in clinical guidelines in the USA, UK, Ireland, and Australia. Inclusion criteria were (Section 2.2): published in English, studies that were published from 2000 to the present that involved Randomized Controlled Trials RCTs, both experimental and non-experimental studies, publications in peer-reviewed journals, including Alzheimer's dementia in any setting and at any level of severity were also utilized. The focus was on treating and managing behavioral and psychological symptoms of dementia (such as agitation, psychosis, and aggression). Pharmacological interventions were considered, and pharmacological interventions from different types were compared or a placebo was used as a comparison. Outcomes of pharmacological interventions and adverse effects were included. Exclusion criteria were (Section 2.2): any study that focused on non-pharmacological interventions only, randomized non-blinded trials, research that was focused on non-dementia populations such as health care workers and caregivers, evidence of low quality, such as case reports, study protocols, commentaries, or design interventions, a population under 65 years of age, articles describing another mental disorder (not dementia).

A priori protocol and JBI methodology for systematic reviews were used in the conduct of this systematic review [15]. It is registered with PROSPERO under CRD42022303438.

In addition to assessing the methodological quality, two independent reviewers critically assessed all eligible studies using the standardized critical appraisal instrument for randomized control trials developed by JBI. The McMaster Quality Assessment Scale of Harms (McHarm) was used to assess the quality of selected RCTs that examined pharmacological harms, intended and unintended adverse effects.

It was decided that, if there was a disagreement between the two reviewers, it would be resolved by a discussion with the third reviewer in order to reach a satisfactory compromise. In addition to the first two reviewers, the third reviewer played a vital role in our research in terms of providing feedback on this paper, suggesting improvements, and making recommendations regarding how to improve the quality of the manuscript, as well as pointing out any errors which needed to be corrected before submission and publication.

In regards to assessing the degree of certainty in the findings, the GRADE approach (Grading of Recommendations, Assessment, Development and Evaluation) was used. In addition, SoFs (Summary of Findings) were designed to provide information on absolute risk, relative risk, and quality of evidence based on the following factors: bias, heterogeneity, directness, precision, and publication bias risks by reviewing the results and outcomes of each selected study.

*2.2. Inclusion and Exclusion Criteria*

| Inclusion Criteria | Exclusion Criteria |
|---|---|
| • This article was originally published in English<br>• Research involving randomized controlled trials, both experimental and non-experimental design<br>• The article was published in a peer-reviewed journal<br>• Inclusion was made of all levels of severity of dementia regardless of the setting in which it oc-curred<br>• Specifically focused on the treatment and man-agement of behavioral and psychological symp-toms associated with dementia (including agita-tion, psychosis, aggression, and conduct prob-lems).<br>• Studies that included pharmacological interven-tions, compared different kinds of pharmacolog-ical interventions, or compared placebo with pharmacological interventions<br>• Including outcomes of pharmacological interven-tions or adverse effects associated with the use of those interventions | • Any study that was solely focused on non-pharmacological interventions<br>• Studies focused on non-dementia-affected popu-lations, such as health professionals and caregiv-ers<br>• Evidence of low quality such as case reports, study protocols, commentaries, or design inter-ventions were not available<br>• The population was younger than 65 years of age.<br>• Papers describing other mental health or psychi-atric disorders (not including dementia). |

*2.3. Intervention*

As part of this study, all pharmacological interventions of antipsychotics, regardless of the dosage and frequency of administration, were considered for inclusion, as well as any studies that assessed whether or not antipsychotics were effective in treating agitation and/or psychosis in the elderly. The study involved hospitals, outpatient clinics, and residential aged care facilities, among others.

*2.4. Comparator*

The pharmacological interventions in studies of antipsychotics were compared to one another and the effects on both agitation and/or psychosis were examined. Drug classification, dosage of the medications, and frequency of administration of the medications were not limited by the study.

*2.5. Outcomes*

As part of this review, the following table in Section 2.5 showing studies that covered any number of the following primary and secondary outcomes were considered:

| Primary Outcomes | Secondary Outcomes |
|---|---|
| <ul><li>A measure of the duration of episodes of agitation or psychosis, regardless of the methodology used to measure them</li><li>The severity of psychosis or agitation, regardless of the assessment method or tool used by the cli-nician</li><li>The frequency with which antipsychotics are used by patients with dementia</li><li>The quality of life of patients before and after the administration of antipsychotic drugs</li><li>In terms of duration and severity, patients' ag-gression against family members, caregivers and health-care professionals should be evaluated regardless of the approach used to assess it or the measures used to measure it</li><li>Including, but not limited to, adverse effects re-lated to antipsychotic interventions, regardless of the approach used as a basis for assessment</li></ul> | <ul><li>Need/use for additional medications to manage the agitation and or psychosis, regardless of the approach used to assess the need.</li></ul> |

*2.6. Search Strategy*

The search strategy was designed to find published studies that met the inclusion criteria described in this review. This review was conducted using a three-step search strategy. In the initial stage of the study, only electronic databases (PubMed, OVID, and CINAHL) as well as Google Scholar and dementia websites were consulted, followed by an analysis of the text words contained in the title and abstract. In the second step, all keywords and index terms identified between 2012 and 2022 were analyzed. Following the identification of citations as potential inclusions for additional studies, the reference lists were searched for all citations. Additional searches were conducted by other members of the team in journals relevant to the specialized topic in order to ensure that the search was comprehensive. Review team members (HQ, MS, JC) determined the journals to be reviewed based on the most commonly accessed journals by clinicians in the field of dementia and neurology at national and international levels. It was decided to limit the search period based on the recommendations regarding the safety and efficacy of antipsychotics that were already clearly presented in previous Cochrane reviews and systematic reviews prior to 2012. Appendix A presents full search strategies.

*2.7. Information Sources*

The databases searched were: Embase via Ovid, PubMed, Scopus (Elsevier), Cochrane Central Register of Controlled Trials, CINAHL plus (EBSCOhost), Epistemonikos, and Ovid MEDLINE, Web of science core collection, International pharmaceutical abstracts (ProQuest), MedlinePlus, and PsychINFO. A web search was performed for primary studies used in relevant clinical guidelines in USA, UK, Ireland and Australia.

*2.8. Study Selection*

Upon finding the citations associated with the search, they were uploaded into End-Note X7 (Clarivate Analytics, Philadelphia, PA, USA) and duplicates were removed. The titles and abstracts of the papers were assessed against the inclusion criteria by two independent reviewers (HQ and JC). In order to assess potential eligibility for inclusion, two independent reviewers retrieved the full text of studies and analyzed it in detail against the inclusion criteria. Studies that met the inclusion criteria were evaluated and entered into the JBI SUMARI (System for the Unified Management, Assessment, and Review of Information), Joanne Briggs Institute, Adelaide, Australia. We excluded studies that did not meet the inclusion criteria. Disagreements between the reviewers (HQ and JC) were resolved through discussion or with the assistance of a third reviewer (MS).

### 2.9. Assessment of Methodological Quality

A standardized critical appraisal instrument for JBI for randomized control trials was used to critically appraise all eligible studies. To assess the quality of other clinical studies that examined pharmacological harms, intended or unintended adverse effects, the McMaster Quality Assessment Scale of Harms (McHarm) was used. In the event of a disagreement between the two reviewers, a third reviewer would resolve the matter through discussion.

### 2.10. Data Extraction

The data for the reviewed studies were extracted by two reviewers using the standardized JBI data extraction tool which was used to extract data from the included studies. It is particularly important to note that the data extracted includes information about the methodology, the intervention types, the populations, the results, and the outcomes of some selected clinical studies. In the event that disagreements arose between the reviewers, they were resolved through a discussion and suggestions and opinions provided by a third reviewer.

### 2.11. Assessing Certainty in the Findings

In order to grade the certainty of evidence, the GRADE approach (Grading of Recommendations, Assessment, Development, and Evaluation) was used. Moreover, the SoF (Summary of Findings) was created with the objective of providing the following information about the absolute risks associated with pharmacological treatments, estimating relative risks, and evaluating the quality of the evidence according to the following criteria: risk of bias, heterogeneity, directness, precision, risk of publication bias by examining the results and outcomes of every study selected.

## 3. Results

A total of 2443 publications were identified, 2403 of which were from electronic databases (PubMed, EBSCOhost, Cochrane library, Ovid MEDLINE, Scopus (Elsevier), BMJ Best Practice, Up to Date). A total of 12 publications were identified from google scholar, 21 from dementia websites (Dementia Centre, Dementia Australia (Australia), Arts for dementia (UK), human brain research–Alzheimer and dementia (USA). Seven hard copy published papers were identified from Neuropsychopharmacology, Clinical Pharmacology and Therapeutics, and American Journal of Psychiatry. A total of 2443 citations were reviewed, of which 312 were excluded due to duplication. Following a review of the title and abstract, 828 citations were excluded (297 were not randomized controlled trials, 31 were not observational nor experimental studies, 417 were not focused on psychotropics in mental health, 61 were not comparable healthcare systems, and 22 were duplicates). The full text reviews of 1303 papers were conducted, and 965 were excluded for not meeting the inclusion criteria (208 papers were not relevant to the inclusion criteria, 53 papers were not about dementia, 511 papers were not about antipsychotics or psychotropics, 129 papers were not about BPSD, and 47 papers were not related to evidence-based practice). A second full text review of 338 papers was conducted, and 286 were excluded (94 studies of lower quality, 88 studies that were not updated, 11 studies focused on non-pharmacological interventions, and 16 studies which did not establish BPSD recommendations). Two independent reviewers (MS and HQ) assessed the remaining 52 studies and agreed to exclude 31 studies due to their lower quality. As a final step, the nine studies were included in the narrative synthesis, which was presented in a Preferred Reporting Items for Systematic Reviews and Meta-analysis flow diagram (PRISMA) in Figure 1. Nine studies were assessed for their methodological quality, and they were subsequently included in the review. An evaluation of the methodological quality of nine studies was conducted in this study.

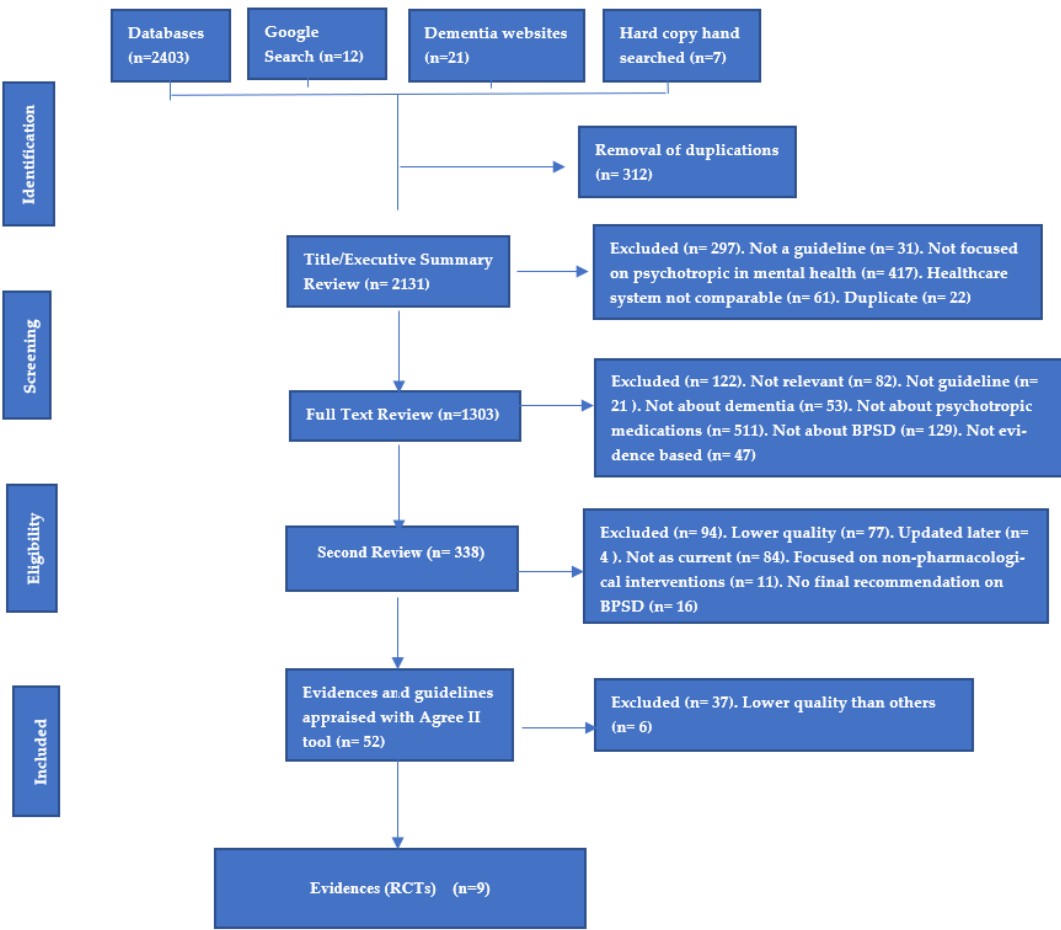

**Figure 1.** PRISMA framework for evidence and guideline review.

*3.1. Methodological Quality*

NINE of the studies included in the review were critically appraised by two independent reviewers (HQ and MS) using the JBI critical appraisal tool for randomized controlled trials (Table 1). In the JBI critical appraisal, there are 13 questions that must be answered. The responses to each question are "no" or "yes" or "unclear".

**Table 1.** JBI Critical appraisal of included randomized controlled trials for antipsychotics.

| RCT Study | Q1 | Q2 | Q3 | Q4 | Q5 | Q6 | Q7 | Q8 | Q9 | Q10 | Q11 | Q12 | Q13 | Total Score |
|---|---|---|---|---|---|---|---|---|---|---|---|---|---|---|
| Tariot et al., 2006 [16] | Y | U | Y | Y | N | Y | N | Y | Y | Y | Y | Y | Y | 10 |
| Sultzer et al., 2008 [17] | Y | U | Y | Y | Y | N | Y | Y | N | Y | Y | N | Y | 9 |
| Kurlan et al., 2007 [18] | Y | Y | Y | Y | Y | Y | Y | U | Y | U | Y | Y | Y | 10 |
| Grossberg et al., 2020 [19] | Y | Y | U | Y | N | N | Y | Y | Y | Y | Y | Y | Y | 10 |
| Brodaty et al., 2005 [20] | Y | Y | U | Y | Y | Y | Y | Y | Y | Y | Y | Y | Y | 12 |
| Ballard et al., 2018 [21] | Y | Y | Y | Y | Y | Y | Y | Y | Y | Y | Y | Y | Y | 13 |
| Ballard et al., 2005 [22] | Y | Y | Y | Y | Y | Y | Y | Y | Y | Y | Y | Y | Y | 13 |
| Mintzer et al., 2007 [23] | Y | Y | U | Y | Y | Y | Y | Y | Y | Y | Y | Y | Y | 11 |
| Streim et al., 2008 [24] | Y | Y | Y | Y | N | U | Y | U | Y | Y | Y | Y | Y | 10 |
| Total Score | 9 | 7 | 6 | 9 | 6 | 6 | 8 | 7 | 8 | 8 | 9 | 8 | 9 | |

Q1. Was true randomization used for assignment of participants to treatment groups?
Q2. Was allocation to treatment groups concealed?
Q3. Were treatment groups similar at the baseline?

Q4. Were participants blind to treatment assignment?

Q5. Were those delivering treatment blind to treatment assignment?

Q6. Were outcome assessors blind to treatment assignment?

Q7. Were treatment groups treated identically other than the intervention of interest?

Q8. Was follow up complete and if not, were differences between groups in terms of their follow up adequately described and analyzed?

Q9. Were participants analyzed in the groups to which they were randomized?

Q10. Were outcomes measured in the same way for treatment groups?

Q11. Were outcomes measured in a reliable way?

Q12. Was appropriate statistical analysis used?

Q13. Was the trial design appropriate, and any deviations from the standard RCT design (individual randomization, parallel groups) accounted for in the conduct and analysis of the trial?

The NINE included studies were critically appraised by two independent reviewers (HQ and MS) utilizing the McMaster Quality Assessment Scale of Harm tool (Table 2). This tool is used to assess risk of harm associated with randomized controlled trials. The McMaster Harm Scale is a tool that consists of 15 questions. As you can see from the table below, the responses to each question across the nine RCT studies are either "no", "yes", or "unclear". The McMaster tool assesses the risk of bias of the included studies. Answering yes means that there is less risk of bias, while answering no implies that the level of bias is quite high. The U letter signifies that the meaning is uncertain.

**Table 2.** Critical appraisal results of eligible antipsychotics RCTs using the McMaster Quality Assessment Scale of Harms (McHarm) critical analysis tool.

| RCT Study | Q1 | Q2 | Q3 | Q4 | Q5 | Q6 | Q7 | Q8 | Q9 | Q10 | Q11 | Q12 | Q13 | Q14 | Q15 | Total Score |
|---|---|---|---|---|---|---|---|---|---|---|---|---|---|---|---|---|
| Tariot et al., 2006 [16] | Y | Y | N | Y | N | N | N | Y | Y | Y | Y | N | N | Y | Y | 9 |
| Sultzer et al., 2008 [17] | Y | Y | Y | N | Y | N | Y | Y | Y | Y | N | Y | Y | Y | Y | 12 |
| Kurlan et al., 2007 [18] | Y | Y | N | N | Y | Y | N | Y | Y | Y | Y | Y | Y | Y | Y | 12 |
| Grossberg et al., 2020 [19] | Y | Y | Y | U | Y | U | U | Y | Y | Y | Y | Y | Y | Y | Y | 12 |
| Brodaty et al., 2005 [20] | Y | Y | Y | U | N | U | Y | Y | Y | Y | Y | U | Y | Y | Y | 11 |
| Ballard et al., 2018 [21] | Y | Y | Y | Y | Y | N | Y | Y | Y | Y | Y | Y | Y | Y | Y | 14 |
| Ballard et al., 2005 [22] | Y | Y | Y | U | N | N | Y | Y | Y | U | U | Y | Y | Y | Y | 10 |
| Mintzer et al., 2007 [23] | Y | Y | Y | U | Y | N | N | Y | Y | Y | Y | Y | Y | Y | Y | 12 |
| Streim et al., 2008 [24] | Y | Y | Y | N | Y | Y | U | Y | Y | Y | Y | Y | Y | Y | Y | 13 |
| Total Score | 9 | 9 | 7 | 2 | 6 | 2 | 4 | 9 | 9 | 8 | 7 | 7 | 8 | 9 | 9 | |

Q1: Were the harms PRE-DEFINED using standardized or precise definitions?

Q2: Were SERIOUS events precisely defined?

Q3: Were SEVERE events precisely defined?

Q4: Were the number of DEATHS in each study group specified OR were the reason(s) for not specifying them given?

Q5: Was the mode of harms collection specified as ACTIVE?

Q6: Was the mode of harms collection specified as PASSIVE?

Q7: Did the study specify WHO collected the harms?

Q8: Did the study specify the TRAINING or BACKGROUND of whomever ascertained the harms?

Q9: Did the study specify the TIMING and FREQUENCY of collection of the harms?

Q10: Did the author(s) use STANDARD scale(s) or checklist(s) for harms collection?

Q11: Did the authors specify if the harms reported encompass ALL the events collected or a selected SAMPLE?

Q12: Was the NUMBER of participants that withdrew or were lost to follow-up specified for each study group?

Q13: Was the TOTAL NUMBER of participants affected by harms specified for each study arm?

Q14: Did the author(s) specify the NUMBER for each TYPE of harmful event for each study group?

Q15: Did the author(s) specify the type of analyses undertaken for harms data?

### 3.2. Characteristics of Included Studies

This systematic review identified nine RCTs: six from the United States, two from Australia, three from the United Kingdom, and two studies conducted in cooperation with international partners, one in conjunction with the United States and one in conjunction with the United Kingdom. There were a variety of study designs identified, including double-blind, randomized, placebo-controlled clinical trials, multicenter randomized, double-blind, placebo-controlled parallel group clinical trials, as well as institutionalized parallel-group, randomized, double-blind, placebo-controlled clinical trials. A total of 458 facilities, including nursing homes, outpatient clinics and medical schools' affiliated hospitals, participated in the nine RCTs. As part of all included RCTs, 2152 older persons, of all genders, were identified as having dementia and reporting psychotic symptoms including visual hallucinations, delusions and/or agitation, as determined by the National Institute for Neurological and Communicative Disorders and Stroke and Alzheimer's Disease and Related Disorders Association. Regarding interventions, this systematic review included nine RCTs in total, including four studies that evaluated quetiapine, two studies with haloperidol, one study with olanzapine, three studies with risperidone, one study with brexpiprazole, one study with pimavanserin, and two studies with aripiprazole.

### 3.3. Review Findings

This systematic review found nine randomized control-trial studies with a total of 2152 older participants. Nine studies evaluated atypical antipsychotics (quetiapine, haloperidol, olanzapine, risperidone, brexpiprazole, pimavanserine and aripiprazole). All of the included studies compared antipsychotics with placebo or with other psychotropics. The participants in the included studies were living in institutions, hospitals, community aged care facilities, nursing homes, or a combination of these settings. With data extracted from the included RCT studies, we analyzed all the outcomes outlined in the inclusion criteria. Antipsychotics are presented in this article for their main outcomes in the treatment of psychosis, agitation, or aggression.

### 3.4. Antipsychotics

Nine trials assessed atypical antipsychotics for agitation and psychosis: four trials for quetiapine [16,18,22,25], two for haloperidol [16,17], one for olanzapine [25], three for risperidone [16,17,20,25,26], one for brexpiprazole [19], one for pimavanserin [21], and two for aripiprazole [23,24]. Table 3 is Summary of included RCTs and their efficacy regarding agitation and psychosis in dementia. Moreover, Table 4 is a characteristics of included RCT studies with antipsychotics.

### 3.5. Findings with Quetiapine

Tariot et al., 2006 concluded from 284 older participants, the mean dose of quetiapine was 96.9 mg daily. The result of the mini-mental state examination score stated no difference was found between quetiapine and other atypical antipsychotics. BPRS agitation factor score was however improved with quetiapine when compared to placebo. No difference was noted between quetiapine and other antipsychotics at the same scale. Similarly, no changes were noted between quetiapine and haloperidol in regards to NPI factors (including agitation factor) [16]. OSES subscale and PSMS total score improved with quetiapine but worsened with haloperidol. Somnolence occurred in 36.2% of participants, and also the lowest prevalence of parkinsonism symptoms compared with other antipsychotics. The result of this trial stated that no significant differences in the measures of efficacy

were observed [16]. Moreover, it indicated that quetiapine was generally well tolerated and did not worsen parkinsonism, although it still results in a decline in the measures of daily functioning [16]. In a study conducted by Kurlan et al., quetiapine was compared to placebo in order to assess the efficacy and tolerability of treating agitation and psychosis in patients with dementia/Alzheimer's disease [18]. A brief psychiatric rating scale was used to assess efficacy from baseline at 10 weeks, while a unified PD rating scale was used to assess tolerability over the course of the trial. According to an ITT analysis (95% confidence interval for the difference in change scores between quetiapine and placebo was $-7.1$ and 2.7, respectively, at the $p = 0.380$ level) [18], a confirmatory analysis showed no differences in change scores between quetiapine and placebo ($p = 0.491$). At $p = 0.53$ and ITT ($p = 0.131$), there was no significant difference in the ADCS–CGIC score [18]. Participants with at least one reported adverse event between quetiapine and placebo were at $p = 0.131$, and this included gastrointestinal disorders ($p = 0.180$), nervous system disorders ($p = 0.080$), and psychiatric disorders ($p = 0.192$) [18]. To determine whether quetiapine can reduce agitation in people with dementia, Ballard et al. conducted randomized controlled trials blinded to clinicians, patients, and outcome assessors [22]. The Cohen–Mansfield Agitation Inventory was used to measure agitation, and the Severe Impairment Battery was used to measure cognitive impairment. In this study, no significant differences were observed between quetiapine and placebo during week 6 and week 12 in terms of improvements in agitation [22]. It was estimated that the average change in the Severe Impairment Battery score from baseline was $-14.6$ points (95% Cl $-25.3$ to $-4.0$) lower than the placebo group (made the condition worse) at week 6 ($p = 0.009$) and $-15.4$ points (95% Cl $-27.0$ to $-3.8$) lower at week 26 ($p = 0.010$). Based on these results, it was found that quetiapine significantly worsened agitation in the quetiapine group [22]. CATIR-AD is a clinical trial that Sultzer et al. designed to assess the effects of atypical antipsychotics on behavioral and psychological symptoms associated with dementia and Alzheimer's disease (including psychosis and agitation) [25]. A daily dose of 25 mg to 50 mg of quetiapine was used in this trial. Based on the results from the baseline to week 12, there was no significant difference in CGIC scores between the antipsychotic groups ($p = 0.141$). There was a 52% increase in the percentage of patients with CGIC scores of 'much improved' to 'very much improved' in the quetiapine group [25]. With regard to BPRS scores for measuring withdrawal depression factors, NPI total scores ($p = 0.097$), BPRS hostile suspiciousness scores ($p = 0.072$), BPRS psychosis scores ($p = 0.352$), BPRS agitation scores ($p = 0.078$), BPRS withdrawal depression scores ($p = 0.987$), BPRS cognitive dysfunction scores ($p = 0.603$), and Cornell Depression Scale scores ($p = 0.841$), the results demonstrated that no clinical outcomes were different between the three antipsychotic treatment groups [25].

**Table 3.** Summary of included RCTs and their efficacy regarding agitation and psychosis in dementia.

| Medication | Total No. of Patients | Effective Dose | Psychosis | Agitation | Indication | Side Effects |
|---|---|---|---|---|---|---|
| Quetiapine | 236 | 100 mg daily | ++ | +++ | Agitation/Psychosis | +++ |
| Haloperidol | 122 | 2 mg daily | + | +++ | Agitation | +++ |
| Olanzapine | 92 | 2.5–5 mg daily | ++/worsened psychosis/Improved suspiciousness and aggressiveness | ++ | Agitation/Psychosis | +++ |
| Brexpiprazole | 703 | 2 mg daily | + | +++ | Specific for Agitation | + |
| Pimavanserin | 181 | 17 mg twice daily/or 34 mg daily at one dose | +++ | + | Specific for psychosis | ++ |
| Aripiprazole | 743 | 10 mg < daily | +++ | ++ | Agitation/Psychosis | + |
| Risperidone | 178 | 0.5–1 mg daily | +++ | ++ | Agitation/Psychosis | ++ |

(Psychosis: + minor benefit, ++ moderate benefits, +++ well-established benefits); (Agitation: + minor benefits, ++ moderate benefits, +++ well-established benefits); Side effects: + less side effects, ++ moderate side effects, +++ highly side effects).

**Table 4.** Characteristics of included RCT studies with antipsychotic.

| Study | Study Design | Country/ Setting | Participants | Characteristics of Participants | Assessment Tools | Interventions | Comparators | Length of Follow-Up | Outcomes |
|---|---|---|---|---|---|---|---|---|---|
| Tariot et al., 2006 [16] | Double-Blinded, Randomized, Placebo-Controlled Clinical Trial | 47 sites of home residents and aged care facilities throughout United States | Quetiapine n = 91, Haloperidol n = 94, Placebo n = 99 | >64 years old, residents in aged care facilities, diagnosed with Alzheimer's dementia | Brief Psychiatric Rating Scale Score (BPRS), Clinical Global Impression of Change (CGI-C), Standardized Mini-Mental State Examination (SMMSE), Multi-dimensional Observation Scale for Elderly (MOSES), and Physical Self-Maintenance Scale (PSMS). | Quetiapine group given 25 mg daily and increased by 25 mg per day for a week, then increased 25 mg every four days to a target dosage of 100 mg daily. Based on clinical responses, quetiapine can be increased to maximum 600 mg daily | Haloperidol group given 0.5 mg daily. Increased by 0.5 mg daily, then increased by 0.5 mg every four days over 14 days. Based on clinical responses, haloperidol can be increased to maximum 12 mg daily | 10 weeks (baseline and ongoing follow-up recorded on week 2, 4, 6, 8, and 10. | Quetiapine, haloperidol and placebo showed improvement in measures of psychosis. No significant difference in the medications. Inconsistent evidence of quetiapine and haloperidol in regards to improvement of agitation. Tolerability better with quetiapine compared with haloperidol. |
| Sultzer et al., 2008 [17] | Double Blinded RCT | 42 clinical sites/hospitals and out-patients clinical centers in United State | 421 enrolled patients were randomized initially to masked treatments with olanzapine n = 100, quetiapine n = 94, risperidone n = 85, and placebo n = 142. | >65 years or older, diagnosed with dementia, Alzheimer's type, reported delusions, hallucinations, agitation or aggression for at least 4 weeks. | Psychiatric and behavioral symptoms, Neuropsychiatric Inventory Questionnaire (NPI-Questionnaire), Brief Psychiatric Rating Scale Score (BPRS), Cornell Scale for Depression in Dementia, AD Cooperative Study-Clinician's Global Impression of Change (CGIC), AD Assessment Scale-Cognitive Subscale (ADAS-Cog) and MMSE, Activities of Daily Living Scale (ADCS-ADL), Dependence Scale, Caregiver Activity Scale, and Alzheimer's Disease Related Quality of Life (ADRQL). | The doses were prepared in low dose to high dose as the following: olanzapine (2.5 mg or 5 mg), quetiapine (25 mg or 50 mg), risperidone (0.5 mg or 1 mg), or placebo. | Comparison between the three medications and with placebo, the comparison at 2:2:2:3 ratio. | 36 weeks. The follow-up and the assessment of week 2, week 4, week 8, week 12, week 24, and week 36 of treatment. | Olanzapine, quetiapine, and risperidone provided some clinical symptoms improvement. The three agents provided efficacy for particular symptoms such as anger, aggression, and paranoid ideas. However, functional abilities such as care needs, quality of life, and the three agents do not appear to improve. No improvement in quality of life, no functional ability improvement, no evidence of cost-effectiveness. Moreover, these agents provided undesirable side effects and adverse effects which depend on individual circumstances and vulnerability to adverse effects. |
| Kurlan et al., 2007 [18] | Multicenter randomized, double-blind, placebo controlled parallel groups clinical trial | 15 participating medical centers in United Sates. 40 patients satisfied the inclusion criteria. | Quetiapine group n = 20, and Placebo group n = 20 | 40 patients involved in study, dementia with Lewy bodies n = 23, Parkinson's disease PD with dementia n = 9, Alzheimer's disease with parkinsonism features n = 8 | Brief Psychiatric Rating Scale BPRS, Unified PD Rating Scale UPDRS, Neuropsychiatric Inventory NPI, Standardized Mini-Mental State Examination MMSE, Clinical Global Impression of Change (ADCS-CGIC), The Motor Examination component of the UPDRS for parkinsonism, and Rochester Movement Disorders Scale for Dementia (R-MDS-D) | Quetiapine began at 25 mg at bedtime, based on discussion between researchers and health professionals, the dose may titrate by 25 mg every 2 days based on efficacy for targeted symptoms and tolerability up to maximum of 150 mg twice daily. | Placebo were tablets that matched the shape of quetiapine tablets in size and color | Ten weeks of trial. the dose of quetiapine may titrate by 25 mg every 2 days based on efficacy for targeted symptoms and tolerability up to maximum 150 mg twice daily. | Quetiapine is well tolerated in dementia patients with parkinsonism. Quetiapine did not worsen parkinsonism. The titration of quetiapine did not show significant benefits in treating agitation or psychosis. |

**Table 4.** *Cont.*

| Study | Study Design | Country/ Setting | Participants | Characteristics of Participants | Assessment Tools | Interventions | Comparators | Length of Follow-Up | Outcomes |
|---|---|---|---|---|---|---|---|---|---|
| Grossberg et al., 2020 [19] | Randomized, Double-Blinded, Placebo-Controlled Trials | Study 1, patients were enrolled by investigators at 81 sites in 7 countries: Russia (29.1% of randomized patients), the United States (27.9%), Ukraine (14.8%), Serbia (12.2%), Croatia (8.5%), Spain (4.4%), and Germany (3.0%). In Study 2, patients were enrolled by investigators at 62 sites in 9 countries: Ukraine (28.9% of randomized patients), the United States (22.6%), Russia (19.3%), Bulgaria (17.8%), Canada (4.8%), France (3.3%), Slovenia (2.2%), the United Kingdom (0.7%), and Finland (0.4%). | Study 1 performed in 81 sites in 7 countries, and Study 2 performed in 62 sites in 9 countries. (Note: Study 1 is 433 randomized, and Study 2 is 270 randomized) | Eligible patients were male or female, aged 55−90 years, with a diagnosis of dementia according to National Institute of Neurological and Communicative Disorders and Stroke and the Alzheimer's Disease and Related Disorders Association | Cohen–Mansfield Agitation Inventory (CMAI) (Total score range: 29−203; higher scores indicate more frequent agitated behaviors), and Clinical Global Impression − Severity of illness (CGI-S) as related to agitation. | Study 1 (fixed dose): brexpiprazole 2 mg/day, brexpiprazole 1 mg/day, or placebo (1:1:1) for 12 weeks. Study 2 (flexible dose): brexpiprazole 0.5−2 mg/day or placebo (1:1) for 12 weeks | Brexpiprazole 0.5 mg daily n = 20. Brexpiprazole 1 mg daily n = 137, Brexpiprazole 2 mg daily n = 140 | The studies each comprised a screening period of up to 42 days, a 12-week double-blind treatment period, and a 30-day post-treatment follow-up period. | In study 1: brexpiprazole 2 mg daily demonstrated significantly greater improvement in CMAI total score from baseline to week 12 than placebo. Brexpiprazole 1 mg daily did not show significant improvement compared to placebo. In study 2: brexpiprazole 0.5–2 mg daily did not show statistical superiority over placebo. In general, brexpiprazole 2 mg daily has potential to be efficacious, safe and well-tolerated in the treatment of agitation in Alzheimer's dementia (AAD) |
| Brodaty et al., 2005 [20] | Randomized double blinded, placebo-controlled trial of risperidone for aggression and psychosis | Multi-centers of aged care facilities and nursing homes in Australia | 93 patients in total randomized in two groups, risperidone group n = 46, and placebo group n = 47 | 93 patients satisfy the inclusion criteria and fulfill BPSD of the psychosis/ aggression of dementia/ Alzheimer's criteria. | Behavioral pathology in Alzheimer's disease (BEHAVE-AD) of psychosis subscale. Clinical Global Impression of Severity (CGI-S), Mini-Mental State Exam (MMSE). | The participants randomized with either a flexible dosage of risperidone (0.25 mg–1 mg daily), or placebo | Placebo were tablets that matched the shape of risperidone tablets in size and color | The follow-up was at regular bases of the baseline week, week 4, week 8 and week 12 (endpoint). | Risperidone is an effective antipsychotic agent for reducing psychosis and agitation and improves global functioning in older people diagnosed with dementia/Alzheimer's disease and reporting with behavioral and psychological disorders. Risperidone demonstrated efficacy to moderate severe psychosis of Alzheimer's disease/dementia. |

**Table 4.** *Cont.*

| Study | Study Design | Country/Setting | Participants | Characteristics of Participants | Assessment Tools | Interventions | Comparators | Length of Follow-Up | Outcomes |
|---|---|---|---|---|---|---|---|---|---|
| Ballard et al., 2018 [21] | Randomized, placebo-controlled, double-blind study | 133 nursing homes were screened across the UK | 181 participants were randomly assigned treatment, pimavanserin n = 90 and placebo n = 91. | Participants of either sex who were aged >50 years, diagnosed of Alzheimer's disease/dementia, and reported psychotic symptoms including visual or auditory hallucinations, delusion and/or agitation. | Mini-Mental Sate Examination (MMSE), Neuropsychiatric Inventory-Nursing Home Version (NPI-NH) Psychosis score. Alzheimer's Disease Cooperative Study Clinical Global Impression of Change (ADCS-CGIC). Cohen–Mansfield Agitation Inventory-Short Form (CMAI-SF). | Pimavanserin initiated at TWO of 17 mg tablet daily or Placebo | Placebo were tablets matching the shape of pimavanserin tablets in size and color | The follow-up to 12 weeks. During the double-blinded period, study visits were performed at baseline and days 15, 29, 43, 64, and 85. The follow-up safety was done by telephone at 4 weeks after the last dose of study medication. | Pimavanserin with two tablets of 17 mg daily, showed efficacy in patients with Alzheimer's disease/dementia and psychosis at the primary endpoint of 6 weeks, with an acceptable tolerability and negative effects condition |
| Ballard et al., 2005 [22] | Double blinded (clinician, patient, outcomes assessor) placebo-controlled trial | Care facilities in the North-East of UK | Three groups randomized: atypical antipsychotic (quetiapine) n = 31, cholinesterase inhibitor (rivastigmine) n = 31, and placebo (double dummy) n = 31 | 93 patients with Alzheimer's disease. Dementia and clinically significant agitation. | Cohen–Mansfield agitation inventory for agitation measures, and Severe Impairment Battery for cognition measures. | The attained dose was 25–50 mg of quetiapine twice daily, | 3–6 mg of rivastigmine twice daily (or less than 12 mg daily between weeks 12 and week 26), and placebo | Analysis was initiated at six weeks follow up (14 quetiapine and 14 rivastigmine and 18 placebo) | Quetiapine has no superior efficacy for treatment agitation compared with placebo. Rivastigmine has no superior efficacy for treatment of agitation compared to placebo. Quetiapine demonstrated significant cognitive decline compared to rivastigmine and placebo |
| Mintzer et al., 2007 [23] | Double-blind, multi-center randomized control trials. | 81 study centers of clinical practice in the United States, Australia, Canada, South Africa, and Argentina. | 487 inpatients admitted into the hospitals with psychosis associated with Alzheimer's disease/dementia were randomized either with aripiprazole or placebo. | Patients enrolled between 55–95 years old, that were diagnosed with Alzheimer's disease/dementia, and reported with psychotic symptoms of delusions and hallucinations, who were living in nursing homes or residential aged care facilities. | NPI-NH Psychosis Subscale score for medication efficacy, Clinical Global Impression–Severity of Illness CGI-S score, BPRS Psychosis Subscale, Score and Total score, CMAI total score, MMSE score, ECGs and signs of extrapyramidal symptoms (EPS). Abnormal Involuntary Movement Scale, and Barnes Akathisia Rating Scale. | Patients were randomized to fixed doses of aripiprazole 2 mg daily, 5 mg daily, or 10 mg daily or placebo for a 10-week period. | Placebos were tablets that matched the shape of aripiprazole tablets in size and color | Patients unable to tolerate acute phase study medication were discontinued from the study. Patients not responding by week 6 to the Clinical Global Impression-Global Improvement CGI-I score, were permitted to discontinue blinded therapy and to begin open-label treatment with aripiprazole through week 10 | Aripiprazole showed efficacy in treating both psychosis symptoms and other BPSD in elderly when compared to placebo. This study suggested that 10 mg daily is an effective dose in this patient population, although some patients may achieve symptom control at 5 mg daily. |

**Table 4.** *Cont.*

| Study | Study Design | Country/ Setting | Participants | Characteristics of Participants | Assessment Tools | Interventions | Comparators | Length of Follow-Up | Outcomes |
|---|---|---|---|---|---|---|---|---|---|
| Steim et al., 2008 [24] | Parallel group, randomized, double-blind, placebo-controlled, fixed dose trial institutionalized. | 35 aged care facilities and nursing homes in the United States. | Patients enrolled aged 55–95 years old, diagnosed with Alzheimer's disease/dementia, and who had psychotic symptoms of delusions or hallucinations at least intermittently for more than a month | A total of 256 participants were randomized into aripiprazole n = 131 or placebo n = 125 for a 10 week trial. | Neuropsychiatric Inventory Nursing Home version Psychosis Score, Clinical Global Impression CGI severity Score, Brief Psychiatric Rating Scale Total, Cohen-Mansfield Agitation Inventory, Cornell Depression Scale Score | Aripiprazole dosing was flexible, started at 2 mg daily with titration to a higher dose of 5 mg, 10 mg and 15 mg daily depending on clinical judgment. The recommended titration schedule was 2 mg daily for one week, then increased to 5 mg daily for 2 weeks, then 10 mg daily for 2 weeks, and then 15 mg daily for the remainder to week 10. Decreases from higher to lower doses were allowed for tolerability only. Participant who could not tolerate 2 mg aripiprazole was dropped from the study. | Placebos were tablets matched to the shape of aripiprazole tablets in size and color | NPI-NH and CGI-S assessments were performed at randomization (baseline), and weeks 1, 2, 3, 4, 6, 8 and 10. The BPRS and CMAI were assessed at baseline every 2 weeks during the study. | Aripiprazole showed no significant difference to placebo in regards to psychosis management. Aripiprazole showed significant superiority compared to placebo in regards to improvement of psychological and behavioral symptoms (including agitation, anxiety, depression). |

### 3.6. Findings with Haloperidol

Two RCTs included in this systematic review tested haloperidol in terms of efficacy and safety for dementia-related psychosis and agitation management. Tariot et al., 2006 tested haloperidol in 284 participants at a mean dose of 1.9 mg. The results of this trial stated that the mean of total BPRS scores improved for all haloperidol and quetiapine groups [16]. No differential benefits were seen between haloperidol and other antipsychotics in regards to the mini-mental state examination score (haloperidol versus quetiapine $p$ =0.875, haloperidol versus placebo $p$ = 0.265) [16]. No difference was found between haloperidol and quetiapine with regards to BPRS agitation factor nor NPI psychosis scores. However, in comparison with quetiapine, haloperidol worsened the BPRS anergia score. MOSES withdrawal subscale and PSMS score worsened with haloperidol versus quetiapine as the $p$ value between haloperidol and quetiapine was significant ($p$ = 0.004). Somnolence was higher in haloperidol compared to quetiapine (occurred 36.2% in haloperidol and only 25.3% in quetiapine). Extrapyramidal side effects were reported more frequently in haloperidol compared to quetiapine (9 versus 32, $p$ = <0.001) [16]. The other RCT conducted by Sultzer et al. tested the efficacy of haloperidol and trazodone in 28 patients with dementia. The scores of CMAI, Delusion Scale, and Ham-D were used as measures of the efficacy of both treatments during the trial time [17]. The results stated that there was no significant difference between haloperidol and antidepressants using CMAI score ($p$ = 0.310), Delusion scale ($p$ = 2.271), or Ham-D score ($p$ = 0.550). Haloperidol did not improve psychosis symptoms ($p$ = 0.210) nor delusional signs ($p$ = 0.461). Haloperidol efficacy against delusion measured using the CMAI score, stated that either baseline delusion scale ($p$ = 0.671), or baseline of Ham-D score ($p$ = 0.30) resulted in no significant difference in managing signs of delusions in psychotic episodes [17].

### 3.7. Findings with Olanzapine

From the included studies in this review, only one RCT conducted by Sultzer et al., 2008, tested olanzapine at 2.5 mg to 5 mg daily with other antipsychotics (quetiapine and risperidone and placebo) for 36 weeks of the trial. The main outcomes of this study stated that olanzapine resulted in greater improvement in patients with psychosis and agitation compared to placebo. These conclusions are based on the improvement scores of the Neuropsychiatry Inventory total score ($p$ = 0.007) (95% Cl −10.8, −1.7), and Clinical Global Impression of Change. When compared with risperidone, olanzapine worsened the scores of the BPRS Psychiatric Rating Scale ($p$ = 0.006), while improving the BPRS Hostile Suspiciousness Factor ($p$ = 0.006, and 95% Cl −0.7, −0.1) [25]. Additionally, olanzapine showed worsening symptoms compared to placebo with the BPRS Withdrawn Depression Factor ($p$ = 0.003), (95% Cl 0.1, 0.5). BPRS Cognitive Dysfunction Factor or the Cornell Depression Scale did not reveal a difference between treatments with either olanzapine or placebo ($p$ = 0.533) (95% Cl −1.5, 0.8). However, the ADCS–ADL scale suggests that olanzapine results in a worsening of functional ability compared to placebo ($p$ < 0.001) [25].

### 3.8. Findings with Brexpiprazole

A RCT performed by Grossberg et al., 2020, evaluated the efficacy of brexpiprazole which was used in different RCTs in seven countries. This kind of RCT is divided into two studies [20]. Study 1 gave brexpiprazole 2 mg daily or 1 mg daily or placebo as fixed doses for 12 weeks. Study 2 gave 0.5 mg to 2 mg daily or placebo as variable doses for 12 weeks. The measures of both studies were based on the Cohen–Mansfield Agitation Inventory (CMAI), and the Clinical Global Impression-Severity of illness (CGI-S) as relates to agitation [20]. The safety of each strength of brexpiprazole was also assessed based on the clinical presentation of each patient. The results of study 1 were categorized into different strengths of the medication dosage. At the dose of 2 mg daily, brexpiprazole showed statistical significance compared to placebo from baseline initiation to week 12, the adjusted mean of the CMAI was −3.77; 95%, confidence interval was −7.38 to −0.17; and $p$ = 0.040. At the dose of 1 mg daily, the outcomes did not demonstrate significant

differences compared to placebo from the beginning to week 12; the adjusted mean was 0.23, 95% Cl −3.4 to 3.86, and ($p = 0.90$). In study 2, the variable doses between 0.5 mg to 2 mg daily did not achieve statistical significance compared to placebo. The outcomes stated a mean of −2.34, 95% Cl −5.34 to 0.82, ($p = 0.150$). However, benefits were noted at the maximum dose of 2 mg daily, the results at this dosage were 95% Cl −8.99 to −1.13, and ($p = 0.012$). In regards to the adverse events of brexpiprazole in this RCT, in study 1, At 2 mg daily, the incidence of events was greater than 5%. These events were headache (9.3% versus placebo 8.1%), insomnia (5.7% versus placebo 4.4%), dizziness (5.7% versus 3% placebo), and urinary tract infection (5% versus 1.5% placebo). In study 2, the adverse events at the doses between 0.5 mg to 2 mg daily were headache (7.6% versus 12.4 placebo), and somnolence (6.1% versus 3.6% placebo) [20].

### 3.9. Findings of Pimavanserine

The efficacy, safety, and tolerability of pimavanserin versus placebo were evaluated in 181 patients with dementia/Alzheimer's disease for 12 weeks by Ballard et al. [26]. According to the results of this trial, the baseline score of NPI-NH psychosis for the pimavanserin group was 9.5 (SD = 4.8) and for the placebo group was 10.0 (SD = 5.6). The correlation between the NPI-NH psychosis score after 6 weeks was 3.76 points for the pimavanserin group and 1.93 points for the placebo group. The mean difference was −1.84 at 95% Cl −3.64 to −0.04 and at ($p = 0.045$). At week 6 from the trial time, participants in the pimavanserin group experienced an average reduction of 39.5% in their NPI-NH psychosis score in comparison to 19.3% in the participants in the placebo group [21]. Responses to the treatment as an improvement were observed in 55% from the pimavanserin group versus 37% for the placebo group ($p = 0.016$). Other measures such as the NPI-NH sleep and NPI-NH agitation/aggression, and CMAI–SF scores were all recorded but no differences were observed. After 12 weeks, the results showed no significant difference between pimavanserin, and placebo. The treatment difference was −0.51 and 95% Cl −2.23 to 1.23 at ($p = 0.561$). During the trial, the main side effects were falls (21% treatment versus 23% placebo), urinary tract infection (22% treatment versus 28% placebo), agitation (21% versus 14% placebo) and discontinuation of trial due to the adverse effects of medications (9% in pimavanserin and 12% on placebo). There were no reports of cognitive decline or motor function decline during the period of the trial in either group [21].

### 3.10. Findings with Aripiprazole

A randomized double-bind, placebo-controlled study was conducted by Streim et al., 2008 to evaluate aripiprazole's effectiveness and safety in treating psychosis in nursing home patients with Alzheimer's disease/dementia [24]. Using the Neuropsychiatric Inventory-Nursing Home Version (NPI-NH) measures, this study indicated that aripiprazole (9 mg daily) did not significantly differ from placebo in terms of mean change from baseline to endpoint. The mean change (SD) from baseline to endpoint for placebo was 4.62 (9.56), the mean change for aripiprazole was 4.53 (9.53) with $p = 0.883$, and Clinical Global Impression (CGI) Severity Score at ANCOVA outcomes of placebo (SD from baseline to the endpoint) −0.62 (9.56), aripiprazole −4.53 (9.23), at $p = 0.198$. On the other hand, there were improvements in measures of the Brief Psychiatric Rating Scale and the Cohen-Mansfield Agitation Inventory at −6.16 (29.11) in placebo, and at −10.25 (25.70) in aripiprazole ($p = 0.03$), and the Cornell Depression Scale score at −0.13 (10.18) for placebo and −1.98 (8.25) for aripiprazole ($p = 0.006$) [24]. Based on these findings, aripiprazole showed clinical benefits for behavioral and psychological symptoms related to agitation, anxiety, and depression. In addition, more significant improvement was noted with $p = 0.05$, on the physician related BPRS total score, as well as symptoms of agitation and anxiety as assessed by the NPI-NH item scores [24]. Mintzer et al. conducted a double-blinded placebo RCT, which evaluated the efficacy and safety of aripiprazole at doses of 2 mg, 5 mg, and 10 mg daily in 487 patients with psychosis associated with Alzheimer's disease/dementia [23]. Neuropsy-

chiatric Inventory-Nursing Home 9 NPI-NH version Psychosis subscale score; Clinical Impression-Severity of Illness (CGI-S); Brief Psychiatric Rating Scale (BPRS) Core and Total, and Cohen–Mansfield Agitation Inventory (CMAI) scores were used between baseline and week 10. A significant improvement in clinical outcomes was seen with a strength of 10 mg compared to other strengths or placebo [23]. In the NPI-NH Psychosis subscale, the SD from baseline to endpoint was −6.87 (8.6) versus −5.13 (10.0), at a significance level of 0.013 (analysis of covariance). The outcomes of the CGI-S were −0.72 (1.8) when compared with −4.17 (21.6), which is significant with $p = 0.031$. A BPRS Total result (SD from baseline to endpoint) of −7.12 (18.4) was compared with a BPRS Core result of −3.07, versus −1.74, and $p = 0.007$. There was a −10.96 (22.6) versus −6.64 (28.6) CMAI result ($p = 0.023$) and 65.0 versus 50.0 NPI-NH Psychosis Response Rate ($p = 0.019$). The measures of CMAI and BPRS scores at week 6, at the dose of 5 mg daily showed significant improvement versus placebo ($p = 0.014$). It is important to note that 5 mg daily did not reach a significant rate at week 8 and week 10 compared to placebo at the same measures [23]. Further, 5 mg daily significantly improved agitation and aggression (SD from baseline to endpoint) by −2.3 (6.4) compared to −1.3 (6.8) and ($p = 0.031$), and anxiety by −1.9 (5.6) compared to −1.1 (6.2) and ($p = 0.038$) when compared to placebo. No difference was found between 2 mg daily and a placebo [23].

### 3.11. Findings with Risperidone

In this systematic review, three RCTs were included which examined the efficacy and safety of risperidone against other antidepressants or antipsychotics or placebos. For 36 weeks, Sultzer et al. measured the effects of risperidone and other atypical antipsychotics on psychological and behavioral symptoms in older people with dementia and Alzheimer's, and evaluated their efficacy against signs of psychosis and agitation [25]. As part of the study, risperidone was given to the participant group at dosages ranging between 0.5 mg and 1 mg for initial treatment. Additionally, the dosage prescribed was adjusted based on clinical judgment. The results indicated that risperidone (0.5 mg or 1 mg daily) showed greater improvement in the Neuropsychiatric Inventory total score ($p = <0.001$); Clinical Global Impression of Change Score ($p = <0.001$); Brief Psychiatric Rating Scale (BPRS) Hostile Suspiciousness Factor ($p = 0.003$), and on BPRS Psychosis Factor ($p = 0.010$). In repeated measures analysis, risperidone was found to be significantly different from placebo in terms of the clinical psychotic symptoms at ($p = 0.003$) and BPRS ($p = 0.007$). BPRS Withdrawal Depression Factor showed significant changes in the risperidone group ($p = 0.012$), and ADCS-ADL showed significant differences in the risperidone group ($p = 0.011$) [25]. A placebo-controlled RCT was conducted by Pollo et al. to test the efficacy of risperidone in treating aggression in older people diagnosed with dementia. According to the results of the study, the mean change at the endpoint in the BEHAVE-AD psychosis subscale with risperidone versus placebo was 5.1 versus 3.3 at ($p = 0.039$), and CGI-C was also significant with risperidone at ($p = 0.001$) [17]. Risperidone administration at flexible doses between 0.25 mg and 1 mg daily showed improvement in both measures in week 2. A total of 59% of participants treated with risperidone improved their psychosis symptoms at the end of the study as compared to 26% of participants treated with placebo [20]. An analysis by Gross et al. in 2007 compared antidepressants with risperidone to treat psychotic symptoms and agitation in dementia patients with at least one moderate to severe symptom of hostility, aggression, agitation, delusion, hallucinations, or suspicious behavior [26]. According to Pollock, risperidone was effective in treating psychosis and citalopram was effective in treating agitation. A 12-week trial was conducted. A Neurobehavioral Rating Scale and a Side Effects Rating Scale were administered weekly [19]. There was no significant difference between citalopram and risperidone in terms of agitation (−12.5% for citalopram and −8.2% for risperidone) or psychosis symptoms (−32.3% for citalopram and −35.2% for risperidone), as both treatments decreased these conditions significantly. Risperidone had a significant increase in adverse effects compared with citalopram; for example sedation decreased with citalopram (26.2%) but increased with risperidone (83.3%). With citalopram,

somnolence was not significantly increased pre 0.42 (0.72) and post 0.3 (0.64), while with risperidone, somnolence was significantly increased pre 0.24 (0.6) and post 0.45 (0.79). Neither citalopram nor risperidone showed significant differences in extrapyramidal side effects, but there was a significant difference in tremors and rigidity pre 0.4 (0.9) and post 0.81 (1.31) in risperidone, and pre 0.32 (0.61) and post 0.6 (0.86) in citalopram) [19].

## 4. Discussion

One of the most challenging aspects of managing agitation in Alzheimer's dementia is carer fatigue and fear. Aggression, both physical and verbal, is a well-documented symptom of Alzheimer's dementia and common to patients managed in both the home environment and in care facilities. An article published in 2014 regarding care of dementia patents in the home environment states that over 20% of carers have been the recipient of physical assault caused by the dementia patient [27]. Physical violence is also seen as one of the main contributing factors for a person being placed into a care facility. The Australian Nursing and Midwifery Federation (ANMF) published a report by RMIT University, surveying aged care staff members in which 93.3% reported experiencing physical violence at the hands of a resident [3–27]. In a statement made to the Royal Commission for Aged Care Reform in Australia, an aged care facility staff member spoke out about resident-to-staff violence occurring regularly within the residential care setting, and the often-inadequate management strategies and staffing available to prevent incidents of violence as part of agitation and psychosis. The submission also talked about attitudes towards resident aggression with staff being expected to accept violent behavior as a normal part of their job [28]. While there are many non-pharmacological strategies to manage patients with a history of violence, poor staffing ratios and fear of being assaulted at work with no support from employers, may result in staff members feeling that early pharmacological intervention is their only choice. Sufferers of Alzheimer's dementia can also be sexually inappropriate and, at times, sexually violent [29]. With 87.5% of aged care staff reporting being sexually harassed at work [28,29]. Adverse sexual behavior can make carers feel uncomfortable interacting with the resident and reluctant to provide care. If a dementia patient has a history of either physical or sexual violence, carers may be driven to make decisions regarding intervention based on fear and the inability to allot sufficient time to non-pharmacological intervention, thereby increasing the use of pharmacological management strategies. In these cases, carers may view pharmacological intervention as their primary defense against being the recipient of abuse. In cases where agitation is severe, pharmacological intervention is still required even when non-pharmacological intervention is used as the initial management strategy. Figure 2 illustrates a systematic approach that has been implemented by the author in order to treat agitation and psychosis in older individuals with dementia by employing pharmacological strategies.

*Quality of This Systematic Review*

In this systematic review, evidence is synthesized from published RCTs that have evaluated the effectiveness of atypical antipsychotics for treating agitation and psychosis in Alzheimer's dementia patients. As part of this review, the available evidence from randomized controlled trials was analyzed and the effectiveness of dementia treatment was rated. As part of our systematic review, we developed rigorous processes for the critical appraisal and synthesis of the multiple types of evidence in order to help clinicians make clinically informed decisions in the field of healthcare. Using two tools to gather qualitative as well as quantitative information about the research enabled us to form a better review with a lower chance of bias as a result of using both tools. An appraisal tool that provides a critical appraisal of systematic reviews is the JBI Critical Appraisal Checklist for Systematic Review (Table 5), and AMSTART 2: a critical appraisal tool for systematic reviews that include randomized control trials of healthcare interventions (Table 6). For each RCT study included in this review, we conducted a team review (HQ as the primary reviewer and MS as the secondary reviewer) that discussed all items in detail using the above appraisal instruments.

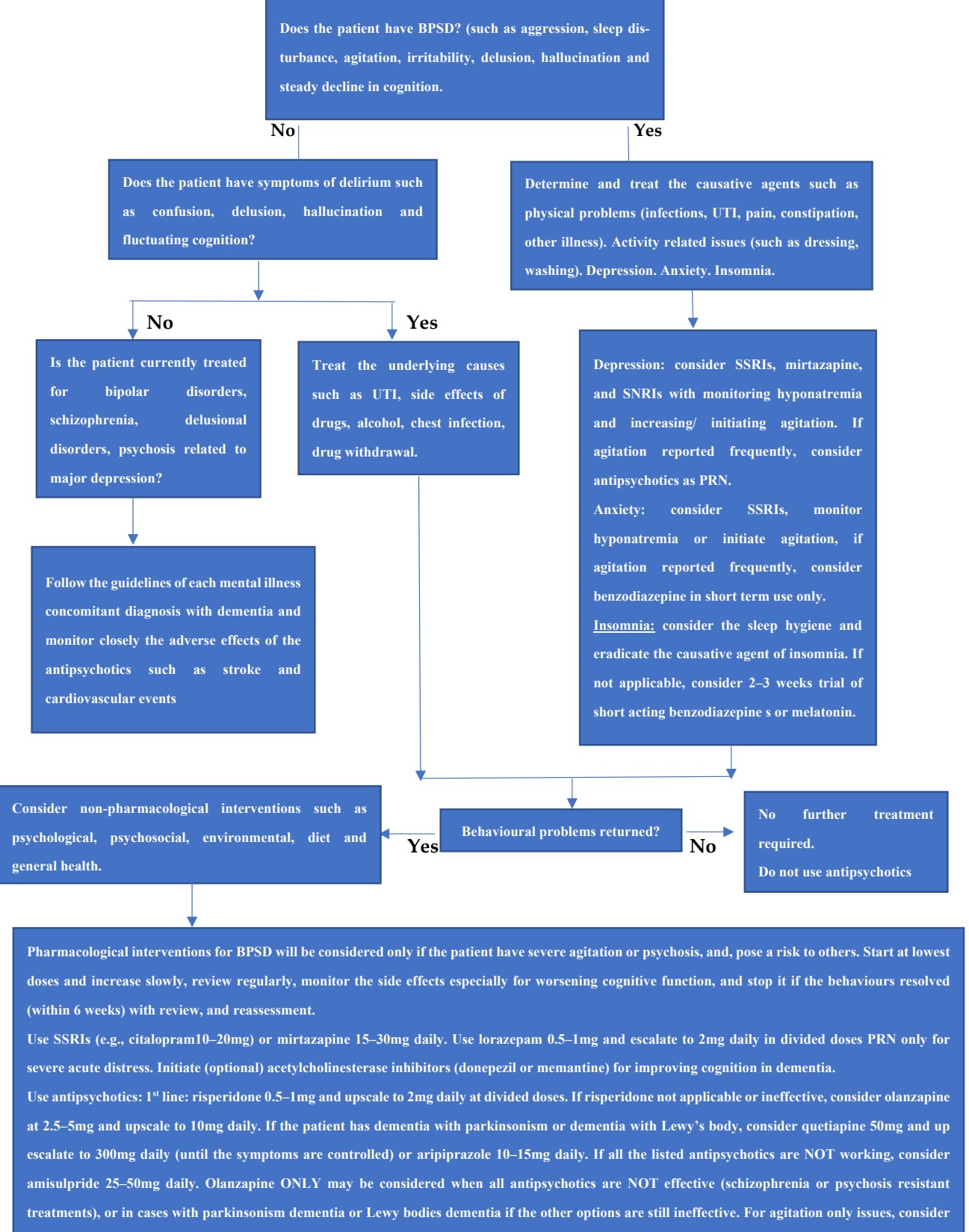

**Figure 2.** Stepwise of psychosis and agitation treatment and management in dementia.

**Table 5.** JBI Critical Appraisal Checklist for Systematic Review.

| | |
|---|---|
| Is the review question clearly and explicitly stated? | Yes |
| Were the inclusion criteria appropriate for the review question? | Yes |
| Was the search strategy appropriate? | Yes |
| Were the sources and resources used to search for studies adequate? | Yes |
| Were the criteria for appraising studies appropriate? | Yes |
| Was critical appraisal conducted by two or more reviewers independently? | Yes |
| Were there methods to minimize errors in data extraction? | Yes |
| Were the methods used to combine studies appropriate? | Yes |
| Was the likelihood of publication bias assessed? | Yes |
| Were recommendations for policy and/or practice supported by the reported data? | Yes |
| Were the specific directives for new research appropriate? | Yes |

**Table 6.** AMSTART 2: a critical appraisal tool for systematic review that included randomized control trials of healthcare interventions.

| | |
|---|---|
| Did the research questions and inclusion criteria for the review include the components of PICO? | Yes |
| Did the report of the review contain an explicit statement that the review methods were established prior to the conduct of the review and did the report justify any significant deviations from the protocol? | Yes |
| Did the review authors explain their selection of the study designs for inclusion in the review? | Yes |
| Did the review authors use a comprehensive literature search strategy? | Yes |
| Did the review authors perform study selection in duplicate? | Yes |
| Did the review authors perform data extraction in duplicate? | Yes |
| Did the review authors provide a list of excluded studies and justify the exclusions? | Yes |
| Did the review authors describe the included studies in adequate detail? | Yes |
| Did the review authors use a satisfactory technique for assessing the risk of bias (RoB) in individual studies that were included in the review? | Yes |
| Did the review authors report on the sources of funding for the studies included in the review? | No |
| If meta-analysis was performed, did the review authors use appropriate methods for statistical combination of results? | N/A |
| If meta-analysis was performed, did the review authors assess the potential impact of RoB in individual studies on the results of the meta-analysis or other evidence synthesis? | N/A |
| Did the review authors account for RoB in individual studies when interpreting/discussing the results of the review? | Yes |
| Did the review authors provide a satisfactory explanation for, and discussion of, any heterogeneity observed in the results of the review? | Yes |
| If they performed quantitative synthesis did the review authors conduct an adequate investigation of publication bias (small study bias) and discuss its likely impact on the results of the review? | N/A |
| Did the review authors report any potential sources of conflict of interest, including any funding they received for conducting the review? | N/A |

In aged care facilities, agitation can cause alarm and concern, especially when it turns into physical aggression associated with behaviors such as screaming, throwing objects, kicking, slamming doors, wandering, biting, pushing, refusal of medications, refusal of assistance, and sexually inappropriate behaviors [30]. In contrast, dementia-related psychosis is the second big issue in aged care facilities [31]. This kind of psychosis is defined by paranoid delusions or intrusive hallucinations occurring after the onset of cognitive decline. Antipsychotics are still effective options to treat agitation and psychosis episodes in dementia [32]. However, before using antipsychotics to treat these conditions, a non-pharmacological option for behavioral symptoms in dementia needs to be considered first,

such as addressing unmet needs such as pain, thirst, hunger, boredom, identification and modification of environmental stressors, behavioral modification, caregiver support, and problem solving [32,33]. Currently, there is no approved pharmacological treatment specifically for either psychosis or agitation in dementia nor a recognized guideline in place. Physicians, as such, are relegated to using unapproved dopamine receptor blocking agents which are normally used for schizophrenia [16,17,19–21,23–34]. New clinical trials are proceeding with several new therapeutic agents in a pathway more specifically targeted either through the psychosis network (e.g., pimavanserin–5HT2A antagonist), or the agitation network (e.g., brexpiprazole–multimodal glutamate and monoamine agents) [16,17,19–21,23–34]. Thus, it is important to distinguish agitation from psychosis as their treatments are directed to entirely different brain networks [34]. In this systematic review, the authors included NINE randomized trials that assessed the effect of one or more antipsychotics with antidepressants or placebo on agitation or psychosis in older people with Alzheimer's dementia. Overall, the review found that there are varying results across the trials and medications that performed well in one trial seemed to underperform or have lowered efficacy in another. This variation of results and efficacy was constant across all the medications reviewed and, while some of it may be due to study design, it should not be ruled out that much of it may be caused by the very nature of dementia. In general, atypical antipsychotics have a modest effect for behavioral and psychological symptoms of dementia (including psychosis and agitation) [35]. This feature needs to be weighed up against the adverse effects of these antipsychotics as they can cause falls, extrapyramidal side effects and reduce quality of life [35]. None of the medications contained in this review are designed specifically for use in dementia. Their primary use is for the treatment of other mental health conditions, with dementia use as a secondary function. Dementia is unlike most psychiatric conditions as it is progressive and relentless, whereas many people with other psychiatric conditions can experience times of normality, dementia destroys the physical structure of the brain making recovery, and remission improbable [36]. In most mental health conditions, once a treatment plan and a dosage are established a patient may be able to remain on that same medication at the same dose for an extended period of time, potentially even gradually lowering the dose if their symptoms remain well managed. This is not the case in dementia, where a person in the early stages of dementia may only need a low dose of medication, that same person in the middle to late stages of dementia may need a greatly increased dose to achieve symptom management, they may also need to switch medications or be on a combination of medications. The disease progression may account for the changes in some results specifically in trials where dosage was set the same for all participants, those who were in early stages of dementia may show strong response, however those in later stages may not respond as well. Potential beneficial effects of symptom reduction in psychosis and agitation with atypical antipsychotic treatment outweigh other undesirable clinical adverse effects which depend on an individual patient's circumstances (including vulnerability to adverse effects, severity of symptoms, and effectiveness toward behavioral interventions) [37]. Treatment with quetiapine, risperidone and haloperidol showed improvement of psychosis without a significant difference between them and showed inconsistent evidence of improvement in agitation. The data provided evidence that haloperidol provides benefits for the delusional symptoms of psychosis, and behavioral improvement, but it provides undesirable side effects. Quetiapine has a tolerability compared with haloperidol, and quetiapine did not worsen parkinsonism, but it does have a significant cognitive decline effect. Moreover, quetiapine should not be used as an alternative treatment to risperidone or olanzapine in people diagnosed with dementia/Alzheimer's disease. Risperidone showed no statistical difference in the treatment of either agitation or psychosis in dementia in comparison with other atypical antipsychotics. In the other RCT it was found to reduce psychosis and improve global functioning in elderly with recorded moderate to severe psychosis in dementia. Risperidone showed some improvement in clinical symptoms including anger, aggression and paranoid ideas but no significant improvement of quality of life nor functioning ability. Aripiprazole

demonstrated comparable improvement in psychotic symptoms with placebo. Additionally, it showed consistent significant benefit in psychological and behavioral symptoms of dementia/Alzheimer's disease such as agitation, anxiety and depression, with the lowest risk of adverse effects compared to other atypical antipsychotics (including extrapyramidal side effects and falls in elderly). Olanzapine improves the signs of psychosis (e.g., hostile suspiciousness, hallucinations, aggression, mistrust, uncooperativeness) and lowers the incidence of agitation. However, it worsens signs of depression and daily functional ability skills. Pimavanserin, a new antipsychotic agent, was trialed and showed efficacy in patients with Alzheimer's disease psychosis. This treatment also showed acceptable tolerability with no negative effects on cognition, and no motor function adverse events. The main side effects reported were urinary tract infection, agitation, and fall. Lastly, brexpiprazole a new antipsychotic agent experimented with in many different countries was concluded to have the potential to be efficacious, safe, and well-tolerated for psychosis symptoms in Alzheimer's disease/dementia. The main side effects of brexpiprazole were headache, insomnia, dizziness, somnolence, and urinary tract infection. Based on the information presented below. The following Table 7 shows a comprehensive listing of the antipsychotics that are used for treating agitation and psychosis associated with dementia. The author of the article has designed an extensive recommendation in Table 7 which illustrates the appropriate use of antipsychotics based on the actions of the receptors and the doses at which they should be administered.

In this systematic review, one of the strongest points was that it identified, analyzed, and summarized the findings of all relevant antipsychotics that were used and recorded in the area of dementia. Each antipsychotic medication is presented along with the benefits and drawbacks for each one. The purpose of the present systematic review is to present a robust assessment of the evidence available and to make it more accessible to decision makers to make informed decisions. Furthermore, it provides clinicians with an overview of the evidence they need to assess the risk–benefit ratio of antipsychotics in dementia patients, as well as an overview of the available evidence. The results of this review will also be used to inform future clinical guidelines and to stimulate further research in the area. In addition, each RCT was analyzed using the JBI critical appraisal procedure, the AMSTART2 procedure, and the McMaster Quality Assessment Scales for Harm assessment based on the criteria of the McMaster Quality Assessment Scales for Harm. Due to the well-established measurement methods used in these assessment tools, the information in this review is of high quality and is highly reliable.

A limitation of this review, is that it can be seen that in order to conduct a thorough analysis of this issue, it may be necessary to include both retrospective and prospective studies regarding the effectiveness and safety of the antipsychotic medications used in elderly care facilities and nursing homes to treat dementia patients. Further, there is the possibility that RCTs may need to be widened and involve a larger population. Additionally, it is possible that it may be necessary to use different healthcare systems in different countries. It could be said that there are not a wide variety of RCTs that look at the impact of antipsychotics.

**Table 7.** A summary of the recommendations of how antipsychotics should be used based on the receptors for action and their doses.

| Antipsychotic Agent | Receptors | What That Means | Recommendation |
|---|---|---|---|
| Quetiapine<br>Light point:<br>it has no motor side effects–therefore it can be used for Parkinson's psychosis. Additionally it has no prolactin elevation.<br>Dark point: increases weight gain and worsens metabolic diseases and cardiovascular adverse effects. | D2 antagonist, H1 antagonist, M1 and α1 antagonist (note: quetiapine has plenty of H1 antihistamine properties) | Enhances sleep, daytime sedation | Improve sleep disturbance in bipolar and unipolar depression, managing anxiety disorders.<br>Quetiapine at 50 mg use for insomnia and sleep disturbance (this dose is small, quetiapine at this dose is full of H1 antihistamine and insufficient in amount for 5HT2c or noradrenaline or dopamine transporters. Therefore, this dose, it is not for antidepressant or antipsychotic use)<br>Quetiapine at 300 mg is used for depression symptoms (sufficient number of 5HT2c, and it can be used as an antidepressant)<br>Quetiapine at 800 mg is used for psychosis symptoms (at this dose, quetiapine has a wide binding profile in regards to dopamine D2, 5HT2c, 5HT1A, 5HT7, 5HT2A, 5HT2c, α receptors and H1. This variety manages psychosis symptoms.<br>Quetiapine is approved for bipolar depression, and acute bipolar manic stage. It can be used in combination with SSRIs or SNRIs in unipolar depression that failed to respond sufficiently to antidepressants only.<br>Quetiapine is approved for both schizophrenia and schizophrenia maintenance. |
| | 5HT1a partial agonist, and 5HT2c, α2, %HT7 antagonist actions | All contribute to mood-improving properties | |
| | H1 antagonist actions in combination with 5HT2c antagonist actions | This combination contributes to weight gain and metabolic issues | |
| Haloperidol<br>It should NOT be given to people who have AF, ECG disturbance, respiratory failure, hyperthyroidism, temperature dysregulation, and people with lower WBCs (agranulocytosis) | Potent D2 antagonist, D3, and α1 adrenergic receptor. | Sedative, highly extrapyramidal side effects, orthostatic hypotension, neuroleptic malignant syndrome, increased risk of CVD and increased stroke, QT intervals and ECG disturbance. Highly anticholinergic effects. Increased metabolic syndrome and worsening diabetes and weight gain, hyperprolactinaemia. Worsening motor effects in parkinsonism. | Approved for acute and chronic psychosis<br>Agitation and psychosis in bipolar disorder<br>Managing acute manic stage<br>It can be given in Tourette and other choreas<br>It can be given to people suffering hallucinations due to alcohol withdrawal syndrome (if diazepam is inadequate to manage the condition) |
| Olanzapine<br>Higher doses can be used for people who have not responded to other antipsychotics/or to lower doses of olanzapine.<br>It should NOT be used for prolactinoma, respiratory failure, hyperthyroidism, uncontrolled diabetes or CVDs | Strong potency for D2 receptor antagonism, H1 and 5HT2A antagonist. | This combination contributes to its efficacy in improving mood and cognitive symptoms. However, it increases weight and worsens metabolic issues such as diabetes, cardiometabolic syndromes and peripheral oedema. | It is approved for managing schizophrenia, and for agitation associated with schizophrenia or bipolar disorder (manic stage), and unipolar disorder.<br>Olanzapine (5HT2c and D2) + fluoxetine (5HT2c) can be used for bipolar depression, and treatment of resistant unipolar depression. However, this combination causes weight gain and metabolic issues.<br>It can be used with lithium OR with valproate for bipolar disorder treatment.<br>Acute and chronic psychosis in schizophrenia<br>Agitation in schizophrenia and acute mania |
| Risperidone<br>Paliperidone is an active metabolite of risperidone (long-term depot injection)<br>Dark side: increases prolactin level even at lower doses, and moderate risk of gaining weight especially with children. | 5HT2A and 5HT7 and A2 antagonist, and D2 antagonism. A1 antagonist | Contributes to the efficacy of depression<br>Contributes to orthostatic hypotension and sedation, blurred vision. | Used for schizophrenia maintenance<br>Bipolar mania/maintenance<br>Used for irritability related to autistic disorder<br>Used for self-harm, or self-injury tantrums<br>Used for quickly changing mood<br>Used (off-label) for treatment of agitation and psychosis associated with dementia<br>Acute and chronic psychosis in schizophrenia |

**Table 7.** *Cont.*

| Antipsychotic Agent | Receptors | What That Means | Recommendation |
|---|---|---|---|
| Aripiprazole<br>Light side: it is well tolerated, has the lowest effect on weight gain, cardiometabolic issues, and reduces prolactin. | 5HT1A, 5HT2A, 5HT1D, 5HT2B, 5HT2C, and 5HT7 are partial agonist to antagonists. | As a partial agonist to antagonist. It has antidepressant actions and helps improve the mood disturbance. | Used for schizophrenia and maintenance.<br>Used for treating agitation in schizophrenia and bipolar disorders (at manic stage).<br>Approved for use irritability in children diagnosed with autism, and Tourette syndrome.<br>Approved for adjunction use with SSRIs or SNRIs for major depression disorders.<br>Used for managing bipolar disorder as a monotherapy.<br>Used in combination with lithium or valproate for managing acute manic stage of bipolar disorder.<br>It is NOT approved for managing bipolar depression. |
| | D2 and D3 partial agonist | Based on these features, aripiprazole is an effective agent in treating schizophrenia/maintenance, and bipolar mania. It has relatively lower side effects and reduces prolactin rather than increases it. | |
| | H1 partial antagonism | Less sedative agent or not generally sedative | |
| | Alpha receptors of 1A, 1B, 2A, 2B, 2C | Orthostatic hypotension, headache, light-headedness, | |
| Brexpiprazole<br>Light side: specific for agitation in dementia, and evidence of causing weight gain or sedative or increase risk of cardiometabolic issues. | It is a serotonin-dopamine-noradrenaline antagonist/partial agonist High potent 5HT2A and 5HT1A partial agonist | Acting as antidepressant, antipsychotic and managing agitation.<br>Based on the higher potency toward alpha 1B and 2C and high potency toward 5HT2A. These properties contribute to antidepressant actions. | Approved for managing agitation in dementia<br>Approved for treatment of schizophrenia<br>Not approved yet for managing acute bipolar mania or bipolar depression.<br>Brexpiprazole can be an adjunction administration with SSRI (e.g., sertraline) for managing PTSD<br>Brexpiprazole augmented with SSRIs/SNRIs to treat unipolar major depression<br>Approved for managing unipolar depression |
| | D2 receptor partial agonism. | Specific alpha-1 actions in particular, gives a unique action for managing agitation and psychosis symptoms. | |
| | Higher potency toward Alpha 1B and Alpha 2C antagonism | This feature reduces propensity to cause motor side effects and akathisia (it can be used in parkinsonism as well). | |
| Pimavanserin<br>Light side: approved for managing psychosis in people with parkinsonism and people with dementia. Less risk of metabolic issues or CVD. | It is the only drug with proven antipsychotic efficacy that does not have D2 antagonism/partial agonist actions. It has a potent 5HT2A antagonism with less 5HT2c antagonism. | Strong potency against 5HT2a and 5HT2c, they improve dopamine release in both depression and the negative symptoms of schizophrenia. Moreover, pimavanserin manages dementia related psychosis by reducing the overactivity of the psychosis network caused by plaques, tangles, Lewy bodies, or stroke. This action is achieved by lowering the normal 5HT2a stimulation to surviving glutamate neurons that have lost their GABA inhibition by neurodegeneration.<br>The 5HT2a antagonism in pimavanserin is approved for managing parkinsonism disease psychosis and their positive symptoms. Additionally for managing dementia-related psychosis of all cases. | Approved for managing psychosis and depression<br>Pimavanserin can be augmented with SSRIs/SNRIs in major depression disorders, and it can be co-administrated with D2/5HT2a/5HT1a agents for managing negative symptoms of schizophrenia.<br>Approved for treatment of psychosis in parkinsonism<br>Approved for managing late-stage psychosis in dementia. |

## 5. Conclusions

Behavioral and psychological disorders are still significant challenges in everyday practice at nursing homes and in hospitals. To date, the precise and useful recommendations about agitation and psychosis treatments are still lacking. In clinical practice, many non-pharmacological options are used for managing agitation and psychosis in Alzheimer's dementia. Unfortunately, these kinds of interventions are not the most ideal management strategies for agitation or psychosis. Therefore, pharmacological interventions for BPSD are still the preferred option. The findings from the nine RCTs of atypical antipsychotics in this systematic review not only showed efficacy for treating agitation and psychosis but also determined the levels of tolerability compared between antipsychotics. The outcome of this review suggests conducting further studies involving more participants from aged care facilities, with a longer trial duration to assess the safety and the efficacy of atypical antipsychotics for managing agitation and psychosis in dementia in long term use. In general, this systematic review is an introduction to establish the need for a guideline/s on how to choose the correct antipsychotics and appropriate dosages for the management of BPSD and establish the importance of safe and conservative use of these medications.

**Author Contributions:** Conceptualization, H.Q., J.L.C. and M.D.S.; methodology, H.Q.; validation, H.Q. and M.D.S.; formal analysis, H.Q.; investigation, H.Q. and M.D.S.; resources, H.Q.; data curation, H.Q., J.L.C. and M.D.S.; writing—original draft preparation, H.Q.; writing—review and editing, H.Q., J.L.C. and M.D.S.; supervision, H.Q. and M.D.S. All authors have read and agreed to the published version of the manuscript.

**Funding:** The authors have no sources of funding or other financial disclosures concerning the above document.

**Institutional Review Board Statement:** Not applicable.

**Informed Consent Statement:** Not applicable.

**Data Availability Statement:** Not applicable.

**Conflicts of Interest:** The authors declare that there are no conflict of interest.

### Appendix A. Search Strategy

| Criteria | Inclusion | Exclusion |
|---|---|---|
| Population | Dementia (any type)<br>Aged from 18 to older<br>Cognitive disorders, memory problems, cognitive impairment, Alzheimer's disease, Lewy bodies dementia, vascular dementia, Parkinson's disease with dementia | Any other psychiatry or mental health disorders |
| Intervention | Pharmacological interventions<br>Any Atypical antipsychotics | Treatments not related to psychotropics<br>Non-pharmacological interventions |
| Comparator | Pharmacology vs. non-pharmacology or vs. placebo<br>Pharmacology vs. usual care | Nil |
| Outcomes | Patient's health outcome<br>Morbidity<br>Mortality<br>Quality of life<br>Efficacy<br>Safely<br>Adverse effects | Outcomes among healthcare or caregivers or health professional staff |
| Settings | Community sectors<br>Aged Care Facilities and Nursing Homes<br>Hospitals<br>Out-patient clinics | Nil |
| Study Designs | Randomized Controlled trials (RCTs), Blinded or double blinded -Placebo Controlled or compared with other psychotropics | Case reports<br>Case studies<br>Opinion reports<br>Commentaries<br>Conference abstracts<br>Thesis/dissertations<br>Letters (with no data) |

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
