# Peer review of "Clinical Trial Studies of Antipsychotics during Symptomatic Presentations of Agitation and/or Psychosis in Alzheimer’s Dementia: A Systematic Review"

_2673-5318, doi:10.3390/psychiatryint4030019_

Round 1
Reviewer 1 Report
In the present paper authors propose a systematic review focusing on the treatment approach (appropriate compounds and dosages) for the management of behavioral and psychological symptoms in Dementia. In their work they ultimately included 9 papers after filtering and pruning an initial pool of 2443 publications. Pruning and filtering were performed following JBI guidelines.
Content-wise there are no major issues as the authors summarize and discuss the findings of the selected papers.
There are, however, a series of issues that should be addressed.
- References: According to the journal other papers, references should be in square brackets: e.g. [1] instead of 1.
In introduction, lane 40. The first reference of the paper is indicated with a 2 followed by 3 (lane 41). References should be enumerated in order of apparition.
Some references for the tools/guidelines used in the review are lacking and should be added (for example lanes 88-90 and lanes 152-153 should cite the jbi checklist for systematic reviews according to jbi site indications: https://jbi.global/critical-appraisal-tools). the ref should be: Aromataris E, Fernandez R, Godfrey C, Holly C, Kahlil H, Tungpunkom P. Summarizing systematic reviews: methodological development, conduct and reporting of an Umbrella review approach. Int J Evid Based Healthc. 2015;13(3):132-40.
Introduction feels a bit fragmented and some sentences need to be better adjusted (eg. lanes 51-52)
Methods should be controlled for consistency (e.g exclusion criteria was stated to be 55 years or younger - lane 87, while on lane 120 it is younger than 65). Also, I suggest to use a table or a box for the part included between lanes 106 to 148 for better reading
Results' presentation is heterogeneous. Some times authors use "(p= xxxx)", then use "p values xxxx", "P= xxx" or "p= xxxx". For consistency, I suggest to use the same style throughout the paper. Also, some numbers should be double-checked (e.g. lane 381 - Delusion scale p=2.27 may be not correct).
from lane 430 authors present data using square brackets (lane 434 and following) and then normal brackets (lane 437 and following). Also, what does the number within the brackets represents? in this section punctuation should also be adjusted.
Tables need a legend to better explain their content (e.g. table 3 what does + represents? whats the difference between +, ++ and +++).
Table 4 may be too wide to be included in the main paper. Authors could include it as supplementary material.
Figure 1 should be adjusted as boxes overlap
Figure 2, at the bottom, a 'Yes' text should be 'No'. also figure 2 need a legends to better explain it.
In author contributions section (lane 634 and following) authors may want to remove the first sentence and adjust the rest of the paragraph.
Other than that i have no other comments/suggestions. I thank authors for their work.
Author Response
- References: According to the journal other papers, references should be in square brackets: e.g. [1] instead of 1.
They fixed now.
- In introduction, lane 40. The first reference of the paper is indicated with a 2 followed by 3 (lane 41). References should be enumerated in order of apparition.
They fixed now and enumerated in order of apparition.
- Some references for the tools/guidelines used in the review are lacking and should be added (for example lanes 88-90 and lanes 152-153 should cite the jbi checklist for systematic reviews according to jbi site indications: https://jbi.global/critical-appraisal-tools). the ref should be: Aromataris E, Fernandez R, Godfrey C, Holly C, Kahlil H, Tungpunkom P. Summarizing systematic reviews: methodological development, conduct and reporting of an Umbrella review approach. Int J Evid Based Healthc. 2015;13(3):132-40.
The references added based on your notes and recommendations. Thanks so much for your guidance.
- Introduction feels a bit fragmented and some sentences need to be better adjusted (eg. lanes 51-52)
Right now, well consolidate and not fragmented.
- Methods should be controlled for consistency (e.g exclusion criteria was stated to be 55 years or younger - lane 87, while on lane 120 it is younger than 65). Also, I suggest to use a table or a box for the part included between lanes 106 to 148 for better reading
The inclusion and exclusion criteria designed inside a table and I fixed them based on your notes.
- Results' presentation is heterogeneous. Some times authors use "(p= xxxx)", then use "p values xxxx", "P= xxx" or "p= xxxx". For consistency, I suggest to use the same style throughout the paper. Also, some numbers should be double-checked (e.g. lane 381 - Delusion scale p=2.27 may be not correct).
The errors of typo are fixed now and all unified.
- from lane 430 authors present data using square brackets (lane 434 and following) and then normal brackets (lane 437 and following). Also, what does the number within the brackets represents? in this section punctuation should also be adjusted.
They typos errors are fixed now. I really appreciated.
- Tables need a legend to better explain their content (e.g. table 3 what does + represents? whats the difference between +, ++ and +++).
Below that table, I write down an explanation of each + meaning. Thanks so much for your notes.
- Table 4 may be too wide to be included in the main paper. Authors could include it as supplementary material.
I redesign and I make the table more acceptable for reading.
- Figure 1 should be adjusted as boxes overlap
Thanks for your notes, I redesign it and there is NO overlap anymore.
- Figure 2, at the bottom, a 'Yes' text should be 'No'. also figure 2 need a legends to better explain it.
I fixed this issue and typos errors. I really appreciated
- In author contributions section (lane 634 and following) authors may want to remove the first sentence and adjust the rest of the paragraph.
I removed all these extra notes and right now is more acceptable. I really appreciate your time and effort.
Reviewer 2 Report
"Congratulations to the authors of this paper. The topic addressed in the review is of great interest. The systematic review is exhaustive in introducing this clinical concern and presenting results. The research design is adequate and methodologically correct. Therefore, I hope the paper will be published as soon as possible."Author Response
Thanks for your comments and you actually make me happy and encourage me to publish more and more in psychiatry and psychopharmacology. I hope the other reviewers accepting and proving my paper for publications.
Regards
Reviewer 3 Report
The authors carried out a systematic review focused on clinical trials of antipsychotics for the treatment of agitation and/or psychosis in patients with Alzheimer's dementia. The paper is well-written and of interest for the journal. However, several minor changes should be recommended before considering it for publication.
ABSTRACT
1- The main introduction of the abstract is really long. I recommend to summarize it. I do not recommend to begin with a question.
2- Is this systematic review following the PRISMA statement? What are the search terms? Please, define them.
3- I recommend to add how many studies were referring to each antipsychotic drug (quetiapine, haloperidol, aripiprazole, risperidone, olanzapine, pimavanserin, brexpiprazole).
4- The main conclusions are not a summary or an extensive reflection of the results. Please, rephrase them.
INTRODUCTION
1- I recommend to start the paper by describing the main symptoms of dementia: cognitive symptoms, behavioral symptoms, depressive and psychotic symptoms... Afterwards, it seems to be relevant to link them with risk of violence.
METHODS
1- Research questions should be carefully described in the methods section, not out of the material and methods.
2- I recommend to divide the methods into several sections: screening and selection process, inclusion/exclusion criteria, assessment of quality of studies/risk of bias, etc.
Data extraction should be described before the assessment of methodological quality.
RESULTS (this should be numbered).
1- The number of studies found for each drug is presented in the results section. Please, refer them into the abstract. Number them into the same order in the results and the abstract.
2- Tables should include abbreviations that should be described at the end.
Author Response
ABSTRACT
1- The main introduction of the abstract is really long. I recommend to summarize it. I do not recommend to begin with a question.
I shorten and summarize the abstract, and I removed and rephrased the begin with question. Thanks for your notes.
2- Is this systematic review following the PRISMA statement? What are the search terms? Please, define them.
Yes, I did follow the PRISMA statement, and I mention this now. Thanks so much
3- I recommend to add how many studies were referring to each antipsychotic drug (quetiapine, haloperidol, aripiprazole, risperidone, olanzapine, pimavanserin, brexpiprazole).
I mention it now and written based on your instructions. Many thanks
4- The main conclusions are not a summary or an extensive reflection of the results. Please, rephrase them.
I rephrased the conclusion based on your instructions.
INTRODUCTION
1- I recommend to start the paper by describing the main symptoms of dementia: cognitive symptoms, behavioral symptoms, depressive and psychotic symptoms... Afterwards, it seems to be relevant to link them with risk of violence.
Yes, I add the main symptoms of dementia: cognitive symptoms and behavioral symptoms and I put all these information in the table.
METHODS
1- Research questions should be carefully described in the methods section, not out of the material and methods.
I correct it according to your instructions Thanks so much.
2- I recommend to divide the methods into several sections: screening and selection process, inclusion/exclusion criteria, assessment of quality of studies/risk of bias, etc.
I did the change based on your instructions. Thanks so much.
Data extraction should be described before the assessment of methodological quality.
I changed that based on your instructions.
RESULTS (this should be numbered).
1- The number of studies found for each drug is presented in the results section. Please, refer them into the abstract. Number them into the same order in the results and the abstract.
The reflection between the abstract and the actual result had done and changed based on your instructions. Thanks so much.
2- Tables should include abbreviations that should be described at the end.
The abbreviations already written in details and stated clearly in the major table. I really appreciate your effort and time to make my paper well-designed.
Reviewer 4 Report
The main aim of this systematic review is to compare pharmacological interventions for psychomotor agitation and psychosis symptoms.
The topic is interesting, but there are some major points to address:
1. in the introduction section, the order of the references is not clear.
2. Why are there two sections for “Methods”? The authors should specify. Was the selection and screening of papers from databases based on RTCs or on other systematic reviews? If yes, the authors should describe better the two issues; if no, the authors should report only a methodology section. Moreover, the authors reported different databases, they should specify.
In addition, what’s the difference between “Characteristics of included studies” (line 307) and “Review Findings” (line 320)? (page 9).
3. The Tables should be better laid out.
4. In the Figure 2 in the first panel above, the authors should correct with BPSD instead of PBSD.
5. The discussion section should start resuming the major results obtained from the systematic review and re-highlighting the added value and the originality of their work.
Moreover, it could be important to discuss the main action mechanisms of the different drugs and their relationship with the symptomatology.
6. Are there some limitations linked to the systematic review? Strengths?
Author Response
- in the introduction section, the order of the references is not clear.
The references are fixed now. Thanks so much for your notes.
- Why are there two sections for “Methods”? The authors should specify. Was the selection and screening of papers from databases based on RTCs or on other systematic reviews? If yes, the authors should describe better the two issues; if no, the authors should report only a methodology section. Moreover, the authors reported different databases, they should specify.
They are unified now based on your instructions and notes, and now they are more clear and well defined. Thanks so much for your notes.
- In addition, what’s the difference between “Characteristics of included studies” (line 307) and “Review Findings” (line 320)? (page 9).
Actually, I revised them, and there is no difference. Therefore, based on your instructions and unified them. Thanks for that.
- The Tables should be better laid out.
It is fixed now. Thanks so much.
- In the Figure 2 in the first panel above, the authors should correct with BPSD instead of PBSD.
It is fixed now. Thanks so much.
- The discussion section should start resuming the major results obtained from the systematic review and re-highlighting the added value and the originality of their work.
I add the information based on your instructions and rephrased now.
- Moreover, it could be important to discuss the main action mechanisms of the different drugs and their relationship with the symptomatology.
I did a comprehensive table related to the discussion part included full details in regards the action of mechanism of each antipsychotic in this review and I connected them to the symptomatology of BPSD.
- Are there some limitations linked to the systematic review? Strengths?
I just written down the strong points and limitations in this review. Thanks so much for your notes
Round 2
Reviewer 4 Report
The authors answered all my questions exhaustively. A beautiful work, congratulations.